# Experimental Study of the Effect of Axial Load on Stress Wave Characteristics of Rock Bolts Using a Non-Destructive Testing Method

**Chuanming Li** [1,2], **Xin Xia** [1,2,*], **Ruimin Feng** [3], **Xiang Gao** [1,2], **Xiao Chen** [1,2], **Gang Lei** [4], **Jiankui Bai** [1,2], **Bochao Nie** [1,2], **Zhengrong Zhang** [1,2] and **Baoyou Zhang** [5]

1   School of Mining Engineering, Anhui University of Science and Technology, Huainan 232001, China
2   Key Laboratory of Coal Mine Safety and Efficiently Caving of Ministry of Education, Anhui University of Science and Technology, Huainan 232001, China
3   Department of Civil Engineering, University of Arkansas, Fayetteville, AR 72701, USA
4   Hunan Sine Electronic Technology Co., Ltd., Xiangtan 411100, China
5   China Coal International Engineering Design and Research Institute Co., Ltd., Beijing 100120, China
*   Correspondence: xinxia0805@126.com; Tel.: +86-189-5547-9858

**Abstract:** Traditional rock bolt inspection methods are destructive and limited. Non-destructive testing (NDT) based on the stress wave method can realize a fast and convenient quality detection of rock bolts. To verify the effectiveness of the stress wave-based anchor NDT method, a multi-functional experimental bench was customized on the basis of a bench-pull tension testing machine. Stress waves can be generated by applying axial loads on rock bolts and then collected using a non-destructive tester. The VMD decomposition method and Hilbert–Huang signal processing method were used to filter and analyze the stress wave signal. The influence of the axial loads of different magnitudes on the stress wave was then investigated. The results showed that the stress wave characteristics of the rock bolt changed with the increase in the axial load. It was found, correspondingly, that the stress wave amplitude decreased gradually and there was a trend of rapid decrease at the beginning and then a slower decline. The change in the time domain amplitude of the stress wave after noise reduction can be used to determine the magnitude of the load on the rock bolt during the elastic deformation stage. Further studies showed that the axial load on rock bolts inversely calculated by the stress-wave time-domain amplitude method is accurate and reliable, which can be validated by comparing the data measured by the rock bolt dynamometer. The research results shed light on the development of the NDT technology on rock bolt inspection, and make this testing method more convenient, efficient, and accurate.

**Keywords:** axial load; stress wave; non-destructive testing; signal processing; time-domain amplitude

## 1. Introduction

The application of rock bolt support technology in coal mining engineering practices has many advantages, such as wide adaptability to various engineering and geological conditions, fast excavating speed, low labor intensity, high support strength, satisfied support effect, and low support cost. In current coal mining engineering, rock bolt support technology has been widely used due to its active control to tunnel rock, which is a representation of the main development direction of future coal mine tunnel support technology [1–3]. However, the quality of anchoring using rock bolts is an essential factor that must be carefully considered, which not only affects the stability of the roadway surrounding the rock, but also highly relates to the safety and efficiency of coal mining. Hence, monitoring the strength evolution of anchoring quality is necessitated. Pullout test, as a traditional technique used to test the pullout force of rock bolts, has been commonly adopted to measure its strength; however, this method is a destructive testing method, and

the pullout force does not fully reflect the anchorage condition of the rock bolt, so it has low adoptability to varying conditions [4,5]. At present, non-destructive testing (NDT) of rock bolts has been attractive in monitoring the anchorage quality of rock bolts, which is of great significance to the safety and efficiency of coal mining [6–8].

The stress wave-based NDT technique of solid anchorage quality has been developed and applied with remarkable technical superiority. It mainly uses the stress wave excited by the exciter to test its propagation characteristics in the rock bolt and then evaluates the anchorage quality. Thurner's research on the nondestructive testing of anchor quality by the stress wave method was performed by the ultrasonic method [9]. Beard studied the optimum excitation frequency of rock bolts at low and high frequencies by using the ultrasonic waveguide method for full-length rock bolts, and concluded that the combination of high and low frequencies can achieve satisfied results for rock bolts up to 3 m in length [10,11]. Charles used the acoustic stress wave method for nondestructive testing of anchor quality. It was concluded that it would be a primary method for nondestructive testing of anchor quality because of its advantages of long measuring distance, low requirements for testing conditions and easy operation [12–14].

Lots of research has been conducted on the propagation law of stress waves in anchor solids. By analyzing the dynamic response of rock bolts under load-bearing conditions, Li et al. concluded that the fundamental frequency and dynamic stiffness of the anchorage system increased as a power function of the load. The dynamic response of rock bolts is significantly related to the relative magnitude of the working load- and ultimate load-bearing capacity of rock bolts [15,16]. Zhang et al. analyzed the propagation velocity of stress waves in anchor sections with different anchorage quality. They found a parabolic relationship between the consolidation wave velocity and the strength of the anchorage medium [17]. Wang studied the wave velocities of stress waves in the free section and anchorage section of rock bolts and reported that the wave velocities in the free section of rock bolts did not vary with the load applied to the rock bolts during the time-domain analysis. The load-basis frequency function of the test signals was presented to be a power-of-three relationship by fitting the load-basis frequency function [18].

The effect of pre-stress was considered by Li et al., and the effect of pre-stress on the vibration frequency and time domain waveform of the rock bolt was also analyzed. As the axial working load (pre-stress) increased, the impact of the end pallets on the detected waveform increased, and the reflection was more substantial at the pallets. In contrast, the peak value of the inverted reflection waveform at the start of anchoring gradually decreased [19,20]. Wan et al. analyzed the propagation law of the excitation stress wave in the pre-stressing anchorage system. The size of the pre-stressing and bonding strength of the anchorage body not only affected the reflected waveform and peak size but also affect the generation of secondary reflections within the anchorage body [21]. Qin found that the stress wave velocity slightly increased with the increase in pulling force, the attenuation of the stress wave amplitude weakened, and the difference between each harmonic amplitude decreased during the pulling process of the anchorage system [22].

Studying stress wave propagation in anchors requires processing and analyzing the detection signal. Liang et al. processed the collected stress wave curves, computed their box dimension and fitted the load-box dimension curve to determine the working load and anchorage quality of rock bolts intuitively [23]. Xu uses the wavelet transform to decompose, noise reduce and reconstruct the test signal, and amplify and extract the reflected signal. However, the analysis results depend mainly on the subjective choice of the wavelet basis [24]. Zhang used Hilbert–Huang and other signal processing methods to analyze the detection signals of various intact and defective anchorage models. The analysis process first eliminated the noise in the detection signals by Butterworth filtering. Then, the Fourier and Hilbert–Huang transform (HHT) analysis was performed sequentially on the noise-cancelled signals [25]. Huang et al. used the empirical mode decomposition method to preprocess the NDT signals and the multiscale entropy method to conduct an in-depth analysis of the reconstructed NDT signals, revealing the mechanism by which the

internal structural features of the anchor rod anchorage system affect the complexity of the detection signals [26].

In summary, the axial load plays a role in the vibration frequency and stress waveform in the time domain of the anchor. It affects the changes in stress wave velocity and amplitude decay, but specific stress wave variation characteristics are affected by the complexity of the coal mining environment. For instance, the NDT data of rock bolts with good anchorage quality cannot accurately determine the location of the bottom of the bolt due to the fast decay of the press wave and the weak reflection. In contrast, for rock bolts with poor anchorage quality, the considerable reflection of the bottom of the rock bolt adds difficulty to data analysis, which leads to the failure to comprehensively analyzing the stress wave propagation law in practice. Axial load is an essential indicator for determining anchor quality by nondestructive testing of rock bolts. Most of the work on nondestructive testing of rock bolts focuses on the effect of static and dynamic loads on stress waves, but little research has been conducted on the impact of the axial load on rock bolts' stress wave propagation characteristics, which is meaningful in the actual engineering application since it is more suitable for determining the working load of rock bolts. Moreover, there are few studies on the quality of rock bolt anchorage using the stress wave method. Therefore, in this study, we generate the axial load by stretching the rock bolt with a lying tension tester. The stress wave signal is then collected with a nondestructive testing instrument to further investigate the influence of the law of axial load on the stress wave characteristics in the rock bolt, shedding light on the application of stress wave-based nondestructive testing of rock bolts.

## 2. Experimental Work

### 2.1. The Test Equipment

As shown in Figure 1, a comprehensive experimental bench was built using the anchorage testing machine (Figure 1b) and load-bearing table (Figure 1c). The SET-PWB-01 wireless rock bolt quality tester was based on WiFi wireless data transmission. The test host machine fed a pulsed high voltage into the roller shaker to generate an alternating magnetic field, which drove the rod in the roller shaker to produce a self-excited vibration. This stress wave detection system can detect the stress wave propagation characteristics in the rock bolt during the rock bolt pulling process. The system consists of a mainframe, acceleration sensor, roller exciter, and a data collection system (Figure 1d).

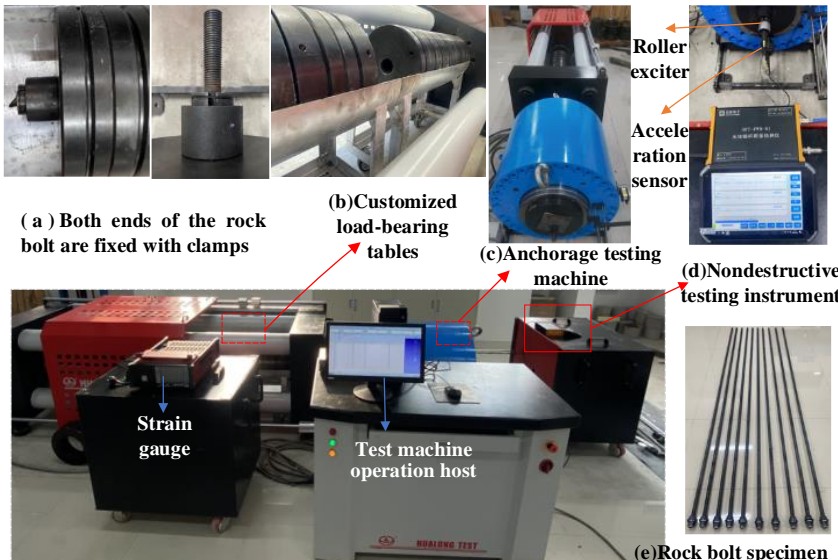

**Figure 1.** Comprehensive test bench for the stress wave-based nondestructive testing technique.

Three 22 mm diameter and 2500 mm length rebar bolts were used as the test specimens. The bolt head was polished to ensure a smooth fit with the sensor and reduce the testing error. The rock bolt specimens (Figure 1e) were fixed with clamps (Figure 1a) at both ends.

### 2.2. The TEST Scheme

The test procedure was determined for the test host machine, and the parameters were also adjusted and set for NDT testing. The signal acquisition parameters were set up mainly to adjust the amplification gain and emission energy to collect the signal of suitable amplitude. The amplification gain is an important means of signal amplification, which is used in conjunction with the emission energy to achieve normal signal acquisition. Amplification gain and emission energy adjustment are used in conjunction with each other—the amplification gain increases with the increase in the signal amplitude. When using the roll exciter, it is not apparent to adjust the amplitude of the time domain signal entirely by increasing the emission energy, so a suitable amplification gain needs to be selected. According to the stress waveform obtained from the test, obvious wave peaks and valleys were observed when the regular time domain signal waveform was selected. In our study, the maximum amplitude of the time domain signal accounted for two thirds of the display box of the full single signal, which was taken as the optimal parameters for the test. Table 1 shows the parameter values used for the study. The axial load was increased from 0 kN to 205 kN, with an increment of 5 kN as a target value. During the loading process, the signal was collected 6 times for each target value until the test host machine judged that the specimen was damaged. The flow chart of the experimental operation is shown in Figure 2.

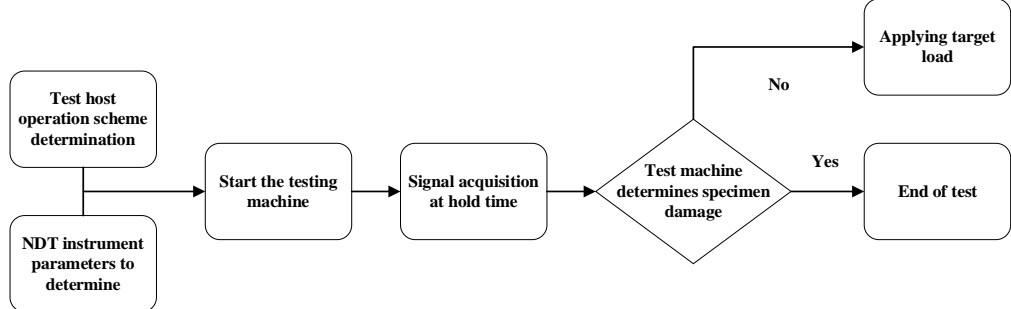

**Figure 2.** Flow chart of experimental operation.

**Table 1.** NDT parameters selected for testing.

| Parameters | Values |
|---|---|
| Sampling rate $f_s$/MHz | 1 |
| Sampling points N/K | 6 |
| Amplification gain L/dB | 62 |
| Emission energy E/J | 10 |
| Trigger threshold $\phi$/(kV/m) | 800 |

It is necessary to keep the tester machine running synchronously with the nondestructive testing system during the test. The tester system automatically collects data including the applied load, displacement, time and the plot of the stress–strain relationship. The stress wave signal collected by the tester can be displayed as a time-domain and frequency signal.

### 3. Analysis of Test Results

#### 3.1. Stress–Strain Analysis under Axial Load

The tensile load–displacement curve of the rock bolt was plotted according to the data output of the testing machine. Two set of rock bolts were tested to verify the reliability of the testing results (Figure 3). From metal fracture mechanics and the law of mild steel tensile stress–strain curve, it can be seen from Figure 3 that the strain increases proportionally with the stress as the load increases before Point A (136 kN) during the pulling process of the rock bolt (Bule curve). The specimen will return to its original state after unloading, so the rock bolt is in the elastic deformation stage. After Point A, a significant displacement can be observed with little changes in the load; it can be deduced that plastic deformation occurs. After Point B, the ability to resist plastic deformation increases again, and the deformation develops faster and increases with the load. The rock bolt is in the strengthening stage; and then necking phenomenon shows up until the ultimate load is finally reached at Point C and the rock bolt fractures. The rock bolts with 12 mm diameter and 1500 mm length also have approximately the same variation pattern (Red curve). Still, due to their different specifications and the different strength of the mild steel itself, the magnitude of the load on the two rock bolts reaching the elastic stage is also different. The focus of the study is on the elastic deformation phase of the rock bolt that occurs during its regular working operation.

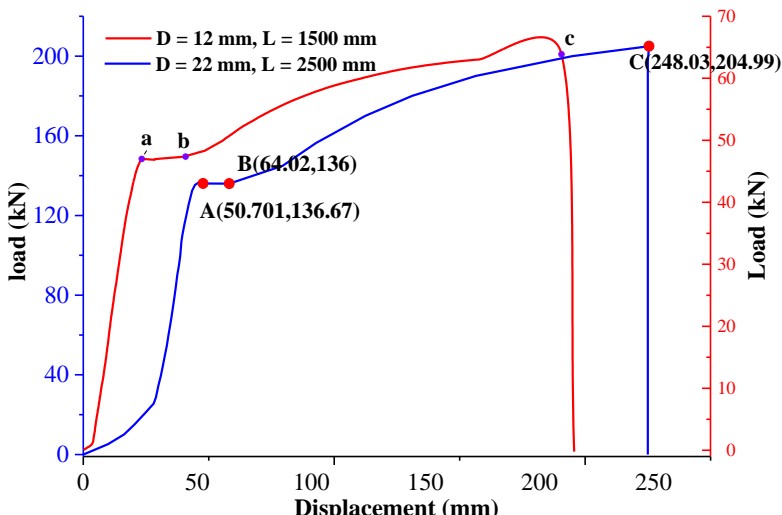

**Figure 3.** Load–displacement characteristic curve of a pullout test on a rock bolt.

#### 3.2. Filtering of Reflected Stress Wave Signal of Rock Bolt Excitation

The detection signal is affected by multiple factors during the data collecting process including the instrument's working noise, Gaussian white noise of the digital-to-analogue conversion, lateral vibration generated by the excitation, and the coupling degree at the receiving end during the acquisition process. Such factors lead to interfering signals in the resulting signal, which make signal identification and analysis difficult. Therefore, noise cancelling is required for the original signal in order to reduce the difficulty and errors during the signal analysis. However, due to the uncertainty of the noise source as mentioned above, it is impossible to avoid the signal noise specifically according to the noise source; hence, signal filtering is an optimal option that can be used to eliminate part of the noise. Firstly, various filtering methods were employed to eliminate the noise in the detection signal, but unfortunately, the filtering effect was not satisfactory [27]. Considering that the collected stress wave signal displayed nonlinear and non-smooth characteristics, such a complex signal can be processed by the empirical mode decomposition (Empirical Mode Decomposition, EMD) [28]. However, the EMD decomposition of the modal components has a more apparent end-point oscillation phenomenon [29], which will make errors in

reducing noise. At the same time, EMD can decompose complex multi-frequency mixed signals into multiple single-component signals containing rich feature information that can be quickly processed. However, the stress wave signal includes many similar time–frequency domain components and noise, resulting in the EMD decomposition of a single signal component containing different frequency components, a phenomenon known as modal aliasing [30,31]. Additionally, EMD is not suitable for processing big data in this study. Therefore, the Variational Mode Decomposition (VMD) algorithm was adopted for noise cancellation [32,33]. This filtering method can improve the endpoint effect and mode aliasing phenomenon and effectively avoid the problems arising from EMD or other signal decomposition algorithms. VMD has been applied to noise reduction processing in the fields of LIDAR signals, monitoring signals, seismic exploration signals, speech signals, etc. [34–37].

The basic principle of VMD is to obtain the best component by solving the constrained variational problem and continuously updating the center frequency and bandwidth of each Intrinsic Mode Function (IMF). The variational model is given in Equation (1) [38,39].

$$\begin{cases} \min\limits_{\{u_k,\omega_k\}} \left\{ \sum_k \left\| \partial_t \left[ \left( \sigma(t) + \frac{j}{\pi t} \right) \otimes u_k(t) \right] e^{-j\omega kt} \right\|_2^2 \right. \\ \qquad\qquad \sum_{k=1}^{K} u_k(t) = f(t) \end{cases} \tag{1}$$

where $u_k$ is the IMF component; $\partial_t$ is the bias operation; $\omega_k$ is the center frequency; $\sigma$ is the unit pulse function; $j$ is the imaginary unit; $K$ is the number of modal decompositions; $\otimes$ is the convolution operation; and *f(t)* is the input signal.

The analytic signal and single-sided spectrum of $u_k$ are first obtained to build this model. Frequency mixing is performed to mix a central frequency for each logical modal signal, and then each modal range is transformed into the entire frequency band. Finally, the parametric number of the squared $L^2$ of the gradient of the demodulated signal is calculated to estimate the modal signal bandwidth [38].

To obtain the extended Lagrange expression, the penalty factor *c* and the Lagrange multiplicative operator $\lambda$ are introduced. The penalty operator alternating direction method is used to receive the optimal solution.

$$\begin{aligned} L(\{u_k\}, \{\omega_k\}, \lambda) = c \sum_{k=1}^{k} \left\| \partial_t \left[ \left( \sigma(t) + \frac{j}{\pi t} \right) \otimes u_k(t) \right] e^{-j\omega_k t} \right\|_2^2 \\ + \left\| f(t) - \sum_{k=1}^{k} u_k(t) \right\|_2^2 + \left\langle \lambda(t), f(t) - \sum_{k=1}^{k} u_k(t) \right\rangle \end{aligned} \tag{2}$$

where $L(``-")$ is the extended Lagrange expression [39,40].

The Fourier isometric transform can be used to complete the adaptive separation of the frequency domain characteristics of the signal. The Fourier inverse transform is then used to convert it to the time domain [41,42].

Figure 4 shows the decomposed intrinsic mode functions of the acquired stress wave time-domain signals, from top to bottom, IMF1, IMF2, IMF3, . . . , IMF8, and the frequency increases gradually from IMF1 to IMF8.

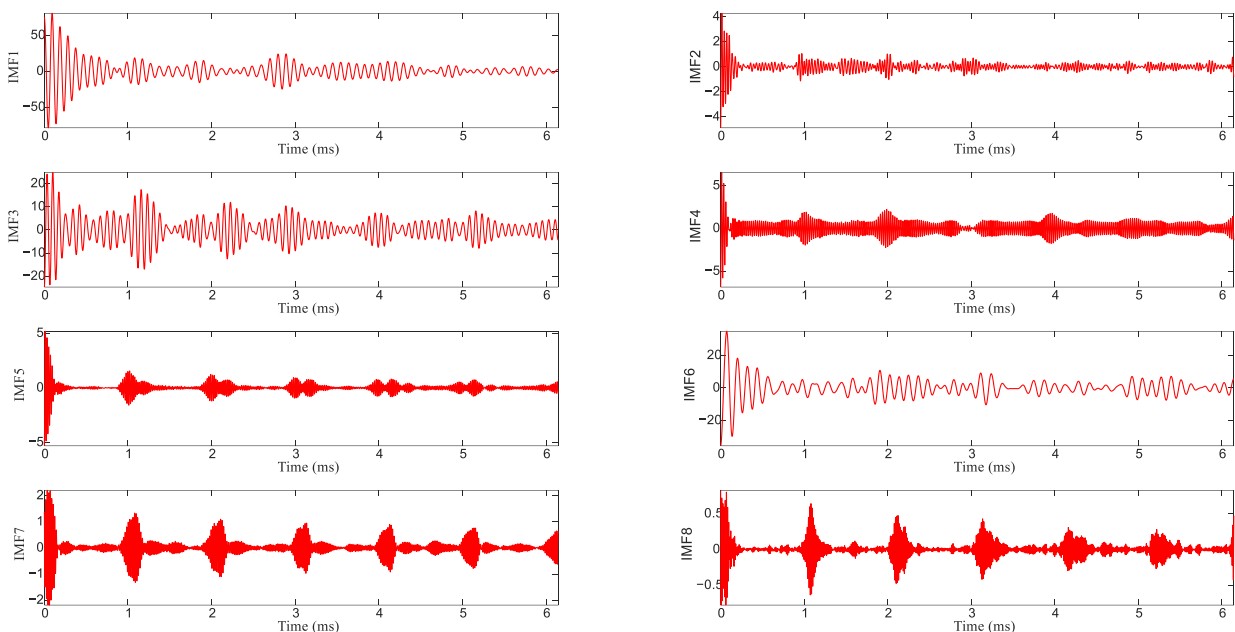

**Figure 4.** Intrinsic mode of a detected signal.

The signal reconstruction is generally performed by removing the high-frequency signal as noise and keeping the low-frequency parts. The noise in the signal can be suppressed by removing the higher-frequency modal components [43]. By removing the higher-frequency modal components, the noise in the signal can be suppressed. From the modal parts in Figure 4, IMF2 to IMF8 can be removed. Only the first modal IMF1 is retained to complete the signal reconstruction and realize the noise reduction process of the original signal. As shown in Figure 5, the measured stress wave signal is filtered to obtain the denoised time and frequency domain plots.

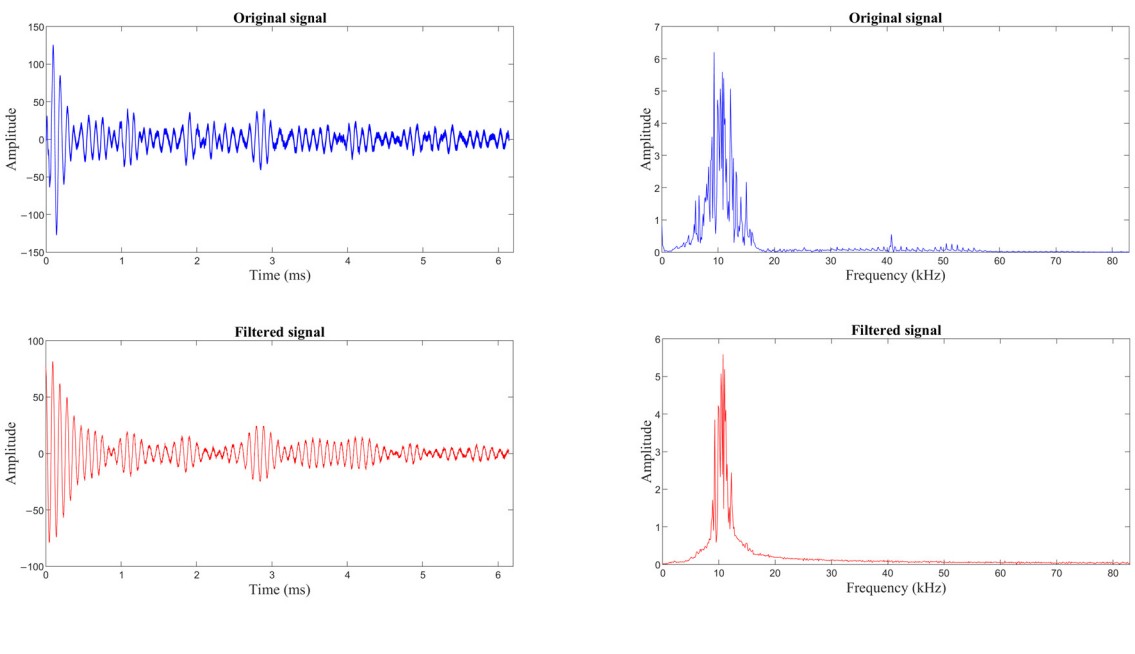

(**a**)

**Figure 5.** *Cont*.

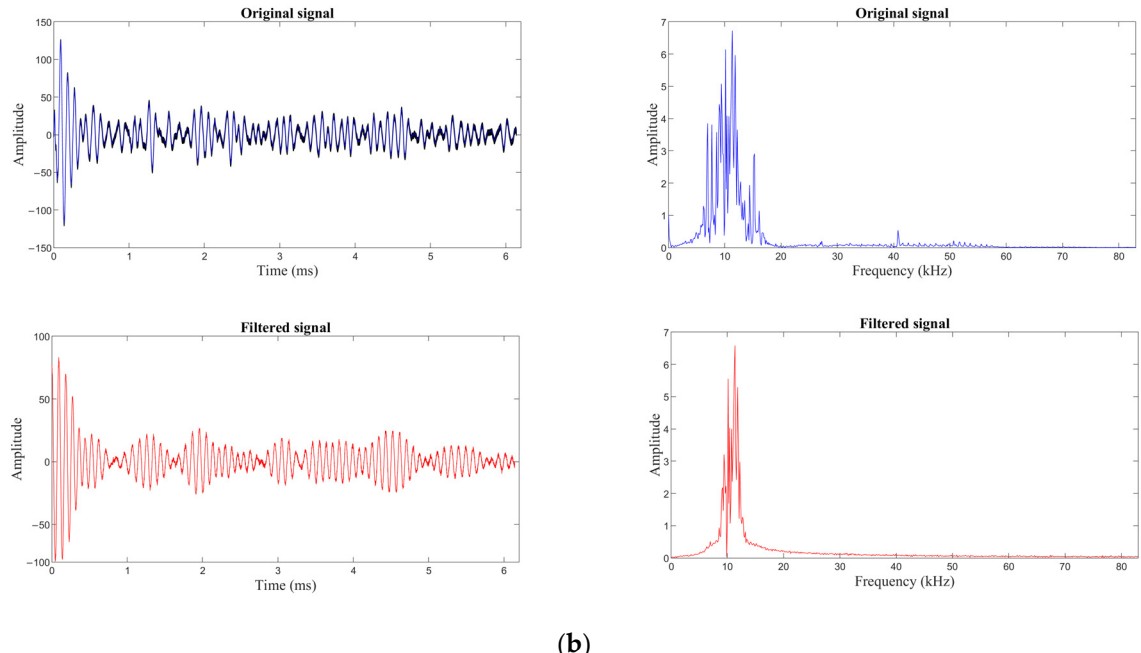

**(b)**

**Figure 5.** Time and frequency domain before and after filtering. (**a**) Time domain waveform diagram. (**b**) Frequency domain waveform diagram.

By comparing the waveforms before and after the noise reduction in Figure 5, the output signal-to-noise ratio improves, and the signal attenuation can be readily seen after the modal noise components are denoised. The decomposition result of VMD avoids the modal mixing problem of EMD decomposition. It does not introduce noise signals in the decomposition, which has a good effect on the processing of non-smooth signals. This verifies that the VMD-based time–frequency analysis method has higher resolution. The results indicate that the method can realize the time–frequency analysis of non-stationary signals, and it is easier to choose an optimal cutoff frequency and retains the nonlinear and non-stationary characteristics of the signal itself. Additionally, the method has an excellent adaptive effect, high operational efficiency and strong noise reduction capability. It can be seen that the VMD denoising method meets our noise reduction requirements and brings a reliable guarantee for the subsequent filtering process.

### 3.3. Time and Frequency Analysis of the Filtered Signal Based on Hilbert-Yellow

Rock bolts are often subjected to a preload during installation, which changes the anchor system when the preload increases to a particular value and thus causes the changes in the characteristics of the stress wave [44,45]. According to "Technical Management Specification for Rock bolt Support in Huainan Mining Area" issued by Huainan Mining Group in 2019, the preload of the rock bolt is required to be 30% to 60% of the rock bolt yield force, i.e., the standard yield strength of rebar rock bolt is greater than 335 MPa, that is, the rock bolt force at yield is at least 127.3 kN. Hence, the anchor preload force is between 38.19 and 76.38 kN. The stress waveform signals at 0 kN, 10 kN, 25 kN, 45 kN, 75 kN, and 100 kN were selected for analysis in this study, and the time domain and frequency domain waveforms of the signals are shown in Figure 6.

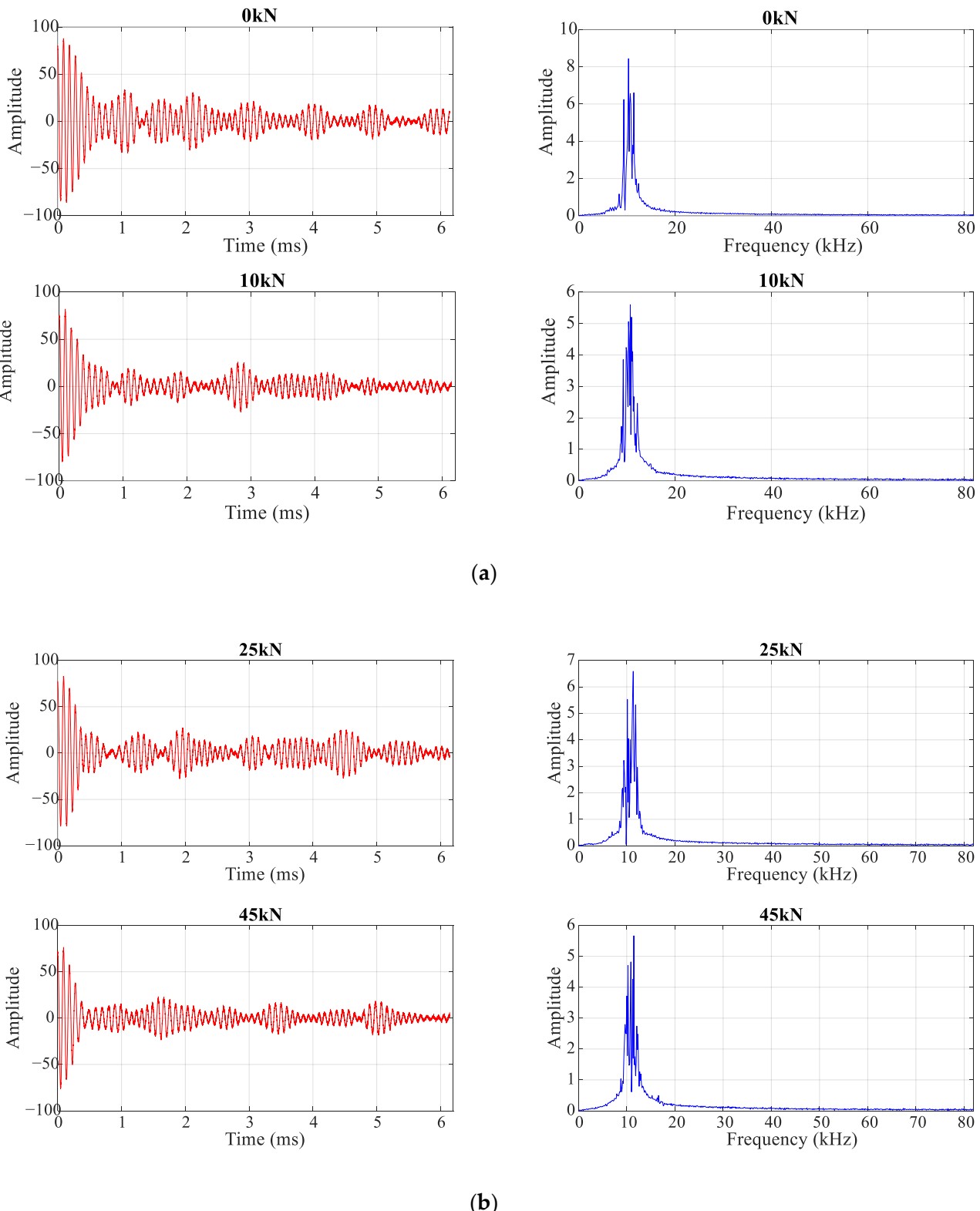

(**a**)

(**b**)

**Figure 6.** *Cont.*

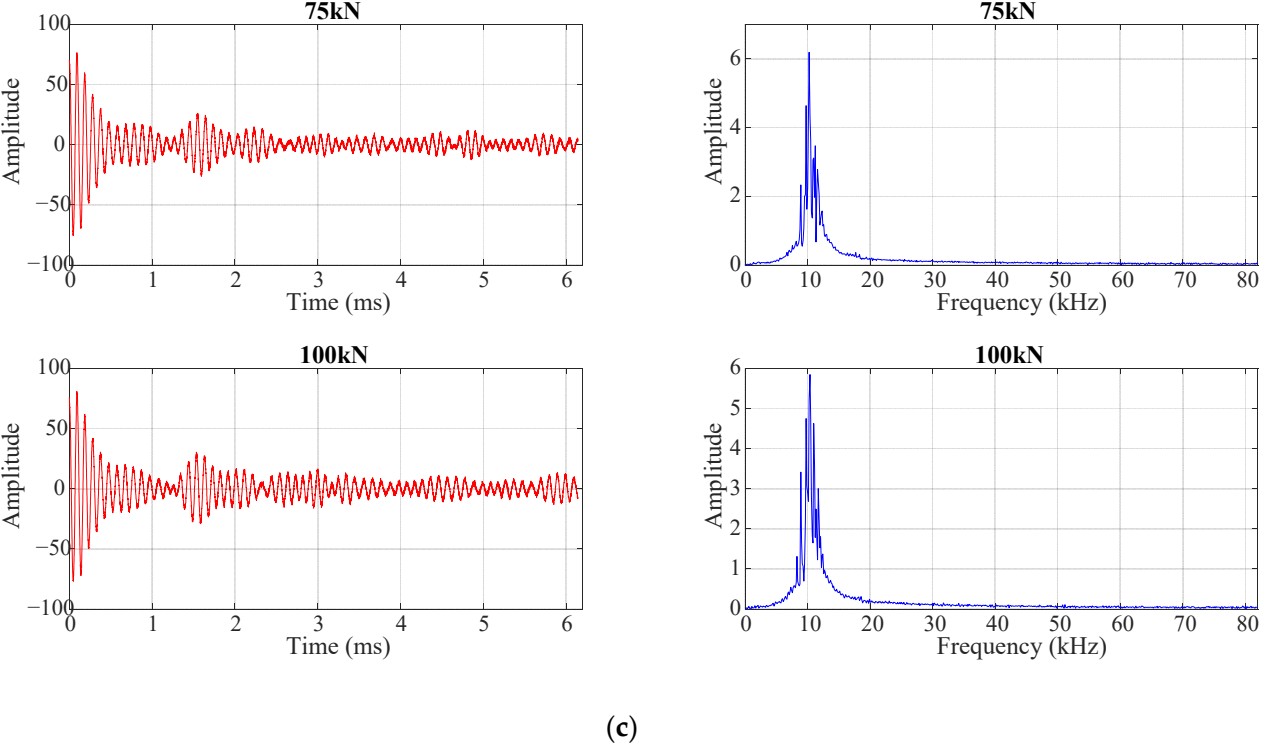

(**c**)

**Figure 6.** Time and frequency domain diagram of NDT signal. (**a**) Time and frequency waveforms of 0 and 10 kN. (**b**) Time and frequency waveforms of 25 and 45 kN. (**c**) Time and frequency waveforms of 75 and 100 kN.

When the stress wave in the rock bolt propagates from the rock bolt head to the bottom end, part of the energy is reflected to form a reflected stress wave due to the impedance at the interface. Since all the tests in this study were performed on intact and well-compacted rock bolts, most energy is transmitted out, and only a tiny portion of the power is reflected. The instantaneous domain signal shows a defect-free reflected wave and a weak or even no reflected wave at the bottom of the rod, making it difficult to study the effect of the axial load on the stress wave in the rock bolt and the phase change in the reflected wave.

From the time domain waveform diagrams in Figure 6, it can be concluded that the stress waveforms for different preloading states are very similar in the time domain during the elastic deformation phase of the rock bolt. The peak values of each cycle wave appear relatively regular. There is no significant difference in the form of fluctuation patterns. For instance, the first wave for all cases shows up in the same time range, i.e., when the load is 0~45 kN, the waveform is regular, with an exponentially fast decaying trend, the amplitude fluctuates within −100~100, and the periodicity of each section of the waveform changes obviously; when the load increases to 75 kN~100 kN, the signal amplitude decreases significantly, and after the end of the first cycle, the waveform changes flatter and flatter, with no obvious decay pattern.

Further analysis of the time-domain waveform changes caused by axial load and common time-domain analysis includes statistical analysis, autocorrelation analysis, and intercorrelation analysis. Standard statistical analysis parameters includes maximum values, RMS, peaks, mean, variance, root mean square, waveform factor, margin, shock index, kurtosis, and skewness [46]. This paper addresses the non-smoothness and non-linearity of the detection signal. Numerical analysis was mainly conducted on the time domain signal by two parameters, root means square and variance, which are summarized in Table 2.

**Table 2.** Time domain waveform characteristics parameters.

| Axial Load (kN) | Root Mean Square | Variance |
|---|---|---|
| 0 | 21.6373 | 468.24695 |
| 5 | 17.1906 | 295.5644 |
| 10 | 17.308 | 299.61435 |
| 25 | 18.984 | 360.45103 |
| 45 | 20.881 | 436.08582 |
| 75 | 23.1176 | 534.50895 |
| 100 | 22.9296 | 525.85211 |

Mean square root, which is the square root of the sum of squares of the signal's sampled values, which is used to characterize the signal's strength and can be calculated by Equation (3).

$$\psi_x(t) = \sqrt{\lim_{N \to \infty} \frac{1}{N} \sum_{i=1}^{N} x_i^2(t)} = \sqrt{E(X^2(t))} \tag{3}$$

The variance, i.e., the square root of the sum of the squares of the deviations of the signal sampling value from the signal means, reflects the degree of dispersion of the data, and the larger the variance, the greater the data fluctuation. The variance can be mathematically expressed by Equation (4).

$$\sigma_x(t) = \sqrt{\lim_{N \to \infty} \frac{1}{N} \sum_{i=1}^{N} [x_i(t) - u_x(t)]^2}$$
$$= \sqrt{E[\{X(t) - u_x(t)\}^2]} = \sqrt{\psi_x^2(t) - u_x^2(t)} \tag{4}$$

As can be seen from Table 2, when the axial load increases from 5 kN to 75 kN, both of the root mean square and variance increase, i.e., the signal intensity of the stress wave of the detection signal increases and fluctuates more violently. As the load continues to increase, the impact of the load on the wave signal intensity becomes smaller. It is known that when the stress state of the main structure changes, the deformation caused by the action of the stress will cause a change in the local or overall stiffness of the structure [47]. When the rock bolt is subjected to tensile stress, the bolt rod is locally deformed elastically, which is reflected by the change in the stress wave signal—that is, the waveform decay is more violent.

The analysis of the amplitude–frequency signal from Figure 6 shows that the Fourier frequency domain waveform shows a similar pattern for all cases: peaking at 10 kHz, showing an asymmetric multi-peak design. These two features make it difficult to extract the characteristic parameters of the frequency domain waveform. Spectral analysis processing of the original data can obtain the signal's spectral content over time and the characteristics of the change in signal intensity. The frequency domain data are filtered using an FFT low-pass filter [48]. The FFT can grasp the global spectral characteristics of the signal. The frequency–domain curves under different loads are plotted in Figure 7.

As can be seen from Figure 7, with the increase in the axial load, the peak amplitude of the stress wave frequency domain decreases and gradually changes from a single-peak form to a multi-peak state, and when the load increases to 45 kN, the peak amplitude appears to "jump" again and returns to a single-peak state, and then the peak amplitude remains, but is asymmetric in multi-peak form. When the rock bolt is subjected to tensile stress, the local stiffness of the rod increases, and the change in stiffness will lead to changes in the resonant frequency of the rod stress wave. The resonant frequency of specific orders will increase. As shown in the stress wave propagation process in the rod, some order peak frequency will increase; for instance, 75 Hz and 100 Hz frequency domain waveforms show that the peak amplitude increases and the resonant peak at the bottom of the rod are apparent.

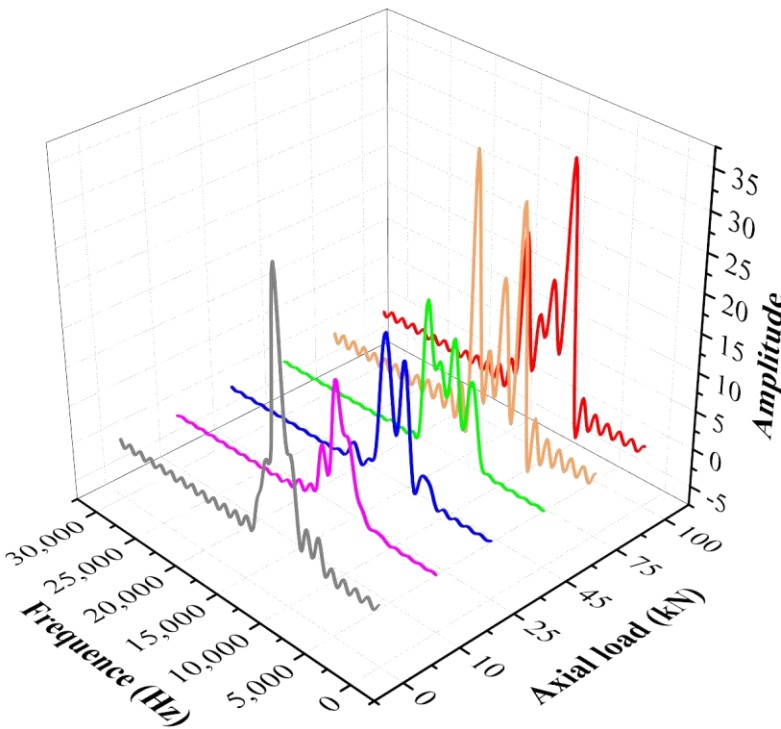

**Figure 7.** Frequency–domain curve of stress waveform for different loads.

From the above analysis, it can be seen that the time domain waveform and Fourier frequency domain of the detection signal cannot directly reflect the variation law of the stress wave characteristics of the rock bolt. The Fourier transform can only obtain the overall distribution of signal frequency but not the relationship between frequency and time, which affects the effect of the Fourier transform on the processing of detection signals. Therefore, the Hilbert–Huang transform (HHT) signal processing method is used to further process the time domain signal [49–52]. The HHT is first to decompose a column of time series data by the empirical modal decomposition algorithm and then obtain the characteristics of this time series data by Hilbert transform.

After the EMD decomposition, HT calculates the instantaneous frequency and amplitude for each IMF component. The impulse response of HT is noted as Equation (5).

$$h(t) = \frac{1}{\pi t}. \tag{5}$$

The HT of the *i*-th IMF is represented as Equation (6).

$$H(IMF_i(t)) = h(t) \otimes IMF_i(t) \tag{6}$$

where $H(``-")$ is used as a function of HT and $\otimes$ is the convolution.

$$Z_i(t) = IMF_i(t) + jH(IMF_i(t)) = a_i(t)e^{j\theta_i i(t)} \tag{7}$$

$$a_i(t) = \sqrt{IMF_i^2(t) + H^2(IMF_i(t))} \tag{8}$$

$$\theta_i(t) = \arctan\frac{H(IMF_i(t))}{IMF_i(t)}. \tag{9}$$

Thus, the instantaneous frequency can be expressed as Equation (10) [53].

$$F_i(t) = \frac{d\theta_i(t)}{dt} . \tag{10}$$

This method is capable of overcoming the shortcomings of the wavelet analysis method and obtaining the signal frequency–time relationship. That is, the signal is preprocessed by VMD and decomposed into basic IMF components. The Hilbert transform is then performed on the basic IMF components. The IMF time–frequency spectrogram and three-dimensional spectrogram of VMD decomposition can be seen in Figures 8 and 9.

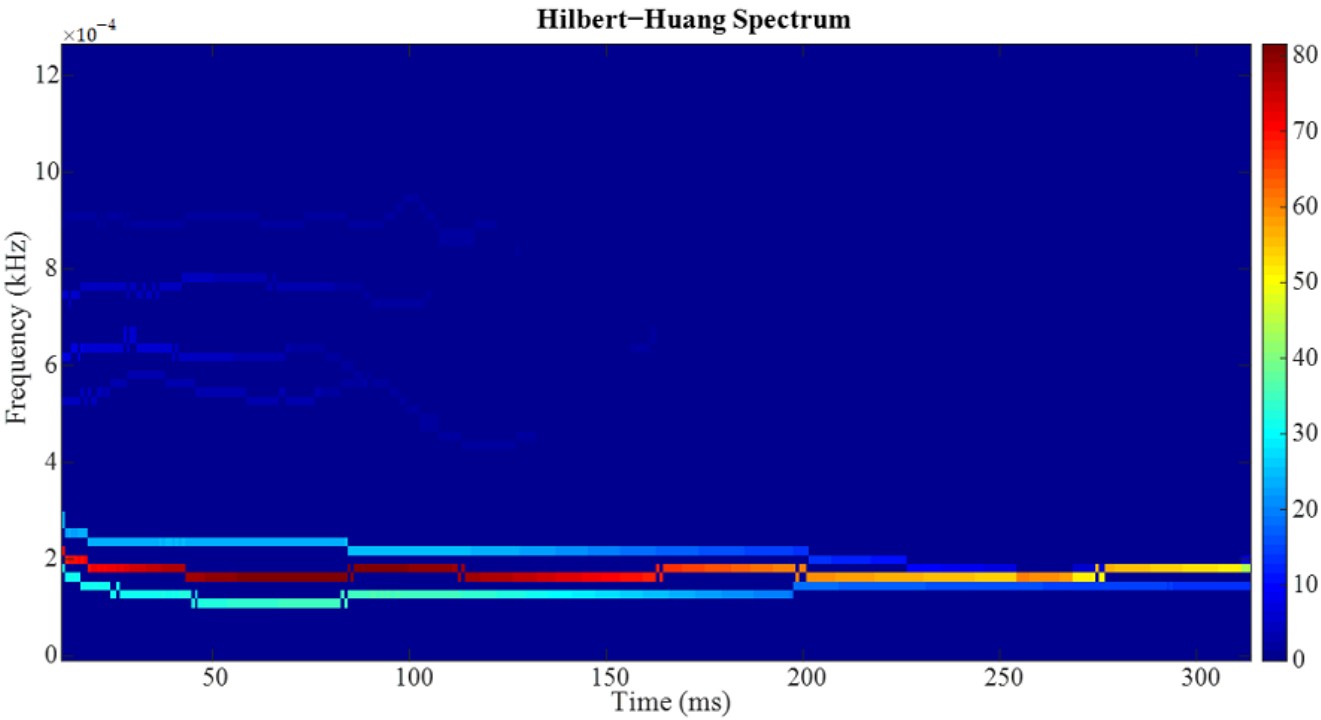

**Figure 8.** VMD algorithm IMF time–frequency spectrogram (10 kN).

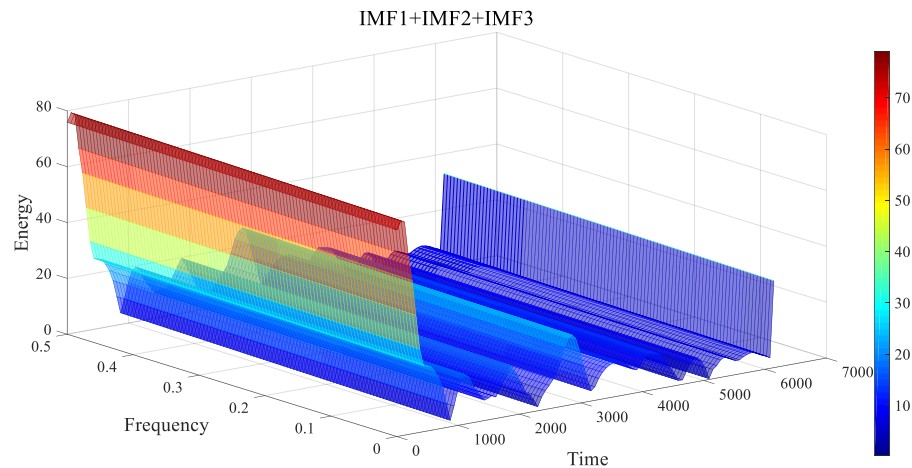

**Figure 9.** 3D spectrum of IMF modal components.

Figure 8 shows the spectra of IMF1, IMF2, IMF3,......, and IMF8 from bottom to top and from low to high frequencies, respectively. The shades of the spectral colors indicate the strength of energy, the power of the first three orders of intrinsic modes IMF1, IMF2 and IMF3 are significant, and the point of the latter methods is relatively small or even negligible. HT can transform the one-dimensional spectral signal into a more information-rich two-dimensional signal and uncover the potential information value of the original data set. It can also be seen from the three-dimensional spectrogram in Figure 9 that the

sum of the first three orders of modal energy occupies the main energy of the signal within 0 to 1 ms (most of the first three orders of modes appear bright red and yellow), and the signal frequency is mainly concentrated in the low frequency, which is consistent with the actual signal characteristics. Therefore, to amplify and capture the signal characteristics without giving away the characteristic signal, IMF1, MF2, and IMF3 are selected as the characteristic intrinsic modes of the original signal for signal reconstruction. Figure 10 shows the time-domain diagram of the reconstructed signal with different loads.

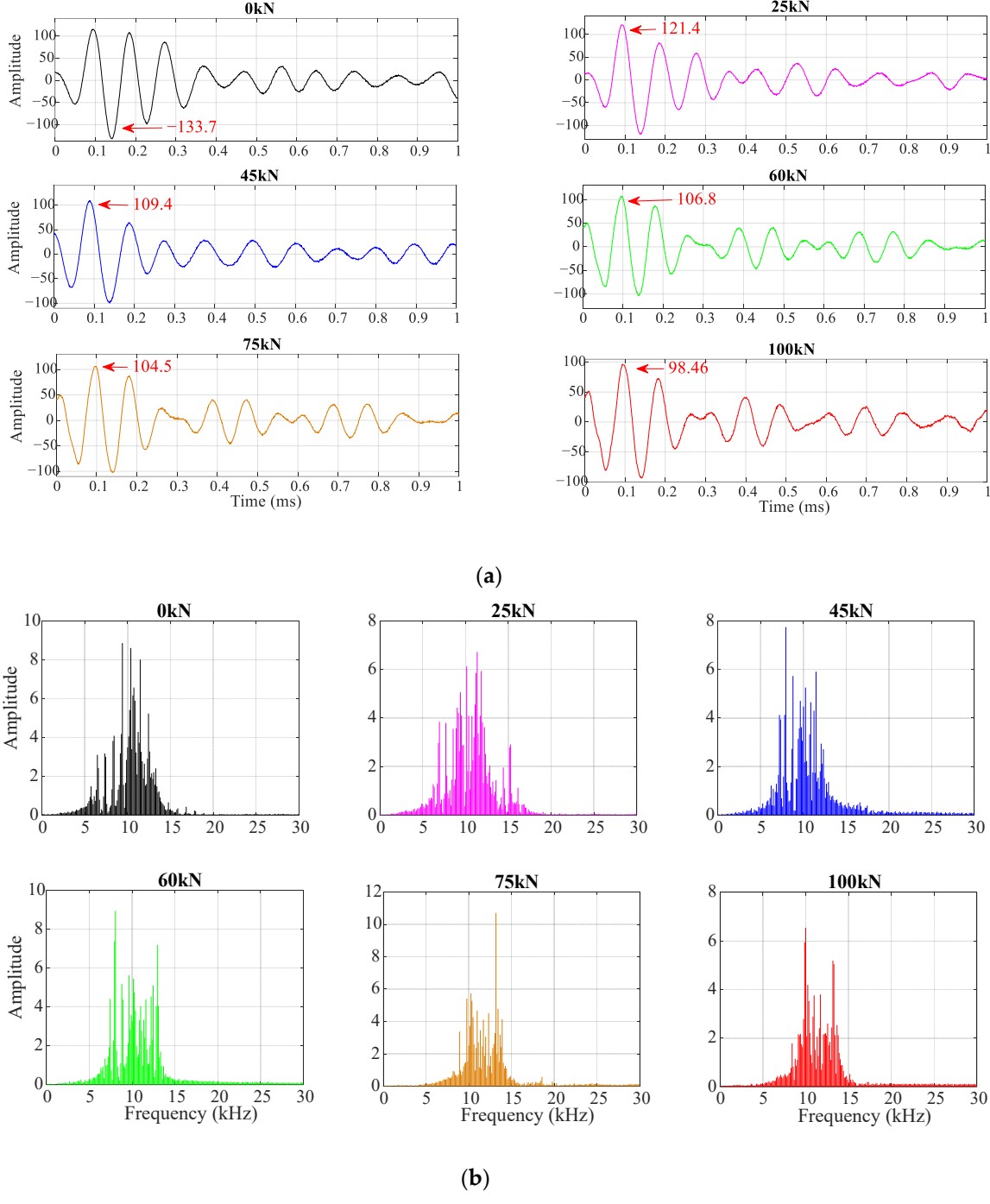

**Figure 10.** Time and frequency domain diagram of reconstructed signal for different loads. (**a**) Reconstructed signal time domain diagram. (**b**) Reconstructed signal frequency domain diagram.

### 3.4. Rock Bolt "Axial Load-Stress Wave Time-Domain Amplitude" Law Analysis

After the above processing, we extracted the reconstructed signal's time domain and frequency domain amplitude under different loads. We obtained the load–amplitude curve as shown in Figure 11. With the increase in the axial load, the stress wave characteristics of the rock bolt vary, and the stress wave amplitude gradually decreases in the time domain, with a trend of rapid decrease and then slow change. It showed a trend of decreasing, then increasing, and then decreasing in the frequency domain. but there was a lack of regularity. The NDT instrument used in this paper uses rod self-excitation technology, i.e., the detector test host generates transient pulses to drive the roll exciter to create alternating magnetic fields. The part of the rock bolt head in the roll exciter generates self-vibration along the rod axial direction under the action of the alternating magnetic field. It is known from the literature that the working load of the anchor rod is a power function of its axial self-vibration frequency (fundamental frequency) of the detected signal under dynamic excitation load [54]—that is, the change in axial load will cause the difference in the stress wave propagation law.

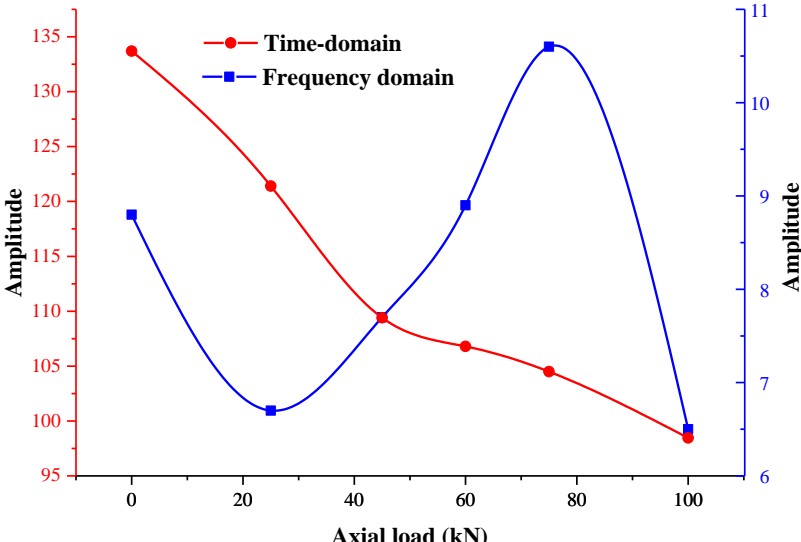

**Figure 11.** Axial load–amplitude curve.

Figure 12 shows a comparison between the load–amplitude curves of three rock bolts with the exact specifications. It can be seen that when the rock bolt is in the elastic deformation stage. With the increase in load, the stress wave time-domain amplitude first decreases rapidly, and then decreases slowly, and the difference of its change trend indicates that the stress wave rock bolt propagates in the process of attenuation drastically. The difference in the trend indicates that the stress wave propagates through the rock bolt with different degrees of attenuation, which indicates that the change in the time domain peak amplitude of the anchor stress wave can characterize the state of the rod in the elastic deformation stage and also verifies the repeatability of the test.

To quantitatively study the effect of axial load on stress wave amplitude, a linear fitting method was used to fit the data, as shown in Figure 13. A good fitting result with $R^2 = 0.935$ can be achieved.

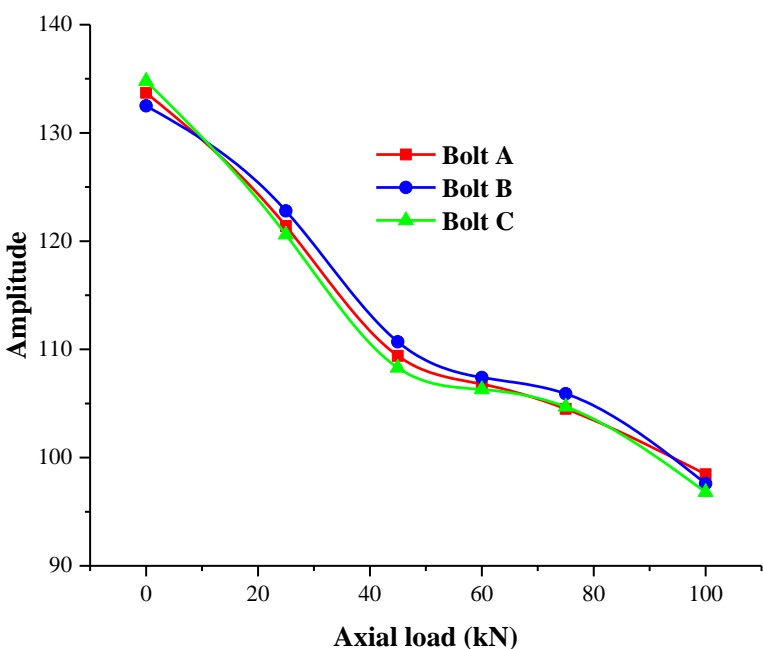

**Figure 12.** Comparison of axial load–amplitude curves.

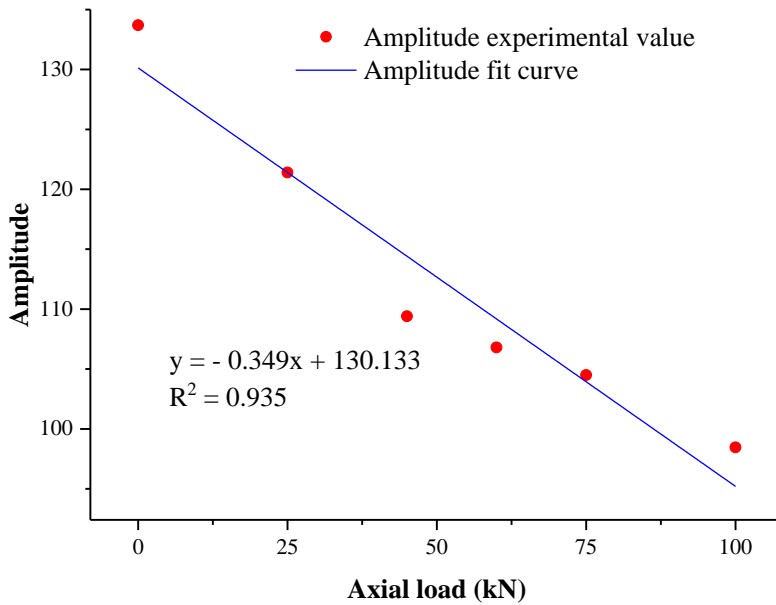

**Figure 13.** Linear fit of axial load–amplitude.

## 4. Experimental Verification of Rock Bolt Axial Load Test Based on Time-Domain Amplitude of Reflected Stress Wave of Excitation

According to the equation of the fitting result to inversely calculate the magnitude of the axial load, and then use the rock bolt dynamometer machine (Figure 14) to determine the axial load of the rock bolt, the axial load of the testing machine loaded to 20 kN, 40 kN, 60 kN, 80 kN were measured and compared with the tensile machine data for analysis (Tables 3–6). Both of the axial loads obtained based on the stress wave time-domain amplitude inverse calculation and measured by the traditional anchor dynamometer method have a certain measurement error, which can be plotted by the results from the two methods against the error in Figures 15 and 16.

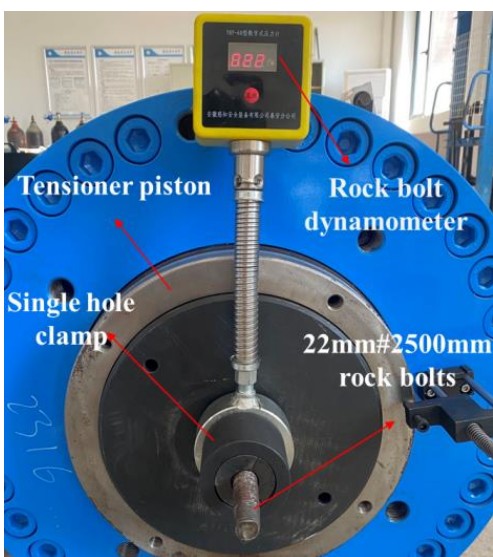

**Figure 14.** Bolt dynamometer testing.

**Table 3.** Measured value of 20 kN axial load applied by the tensioner.

| Test Method | Measured Value (kN) | Measurement Error (Δ/20 kN) |
|---|---|---|
| Rock bolt pulling machine | 20 | 0% |
| Rock bolt dynamometer | 18 | 10% |
| Stress wave method | 19.917 | 0.185% |

**Table 4.** Measured value of 40 kN axial load applied by the tensioner.

| Test Method | Measured Value (kN) | Measurement Error (Δ/40 kN) |
|---|---|---|
| Rock bolt pulling machine | 40 | 0% |
| Rock bolt dynamometer | 37 | 7.5% |
| Stress wave method | 40.035 | 0.193% |

**Table 5.** Measured value of 60 kN axial load applied by the tensioner.

| Test Method | Measured Value (kN) | Measurement Error (Δ/60 kN) |
|---|---|---|
| Rock bolt pulling machine | 60 | 0% |
| Rock bolt dynamometer | 58 | 3.33% |
| Stress wave method | 60.056 | 0.197% |

**Table 6.** Measured value of 80 kN axial load applied by the tensioner.

| Test Method | Measured Value (kN) | Measurement Error (Δ/80 kN) |
|---|---|---|
| Rock bolt pulling machine | 80 | 0% |
| Rock bolt dynamometer | 77 | 3.75% |
| Stress wave method | 80.074 | 0.186% |

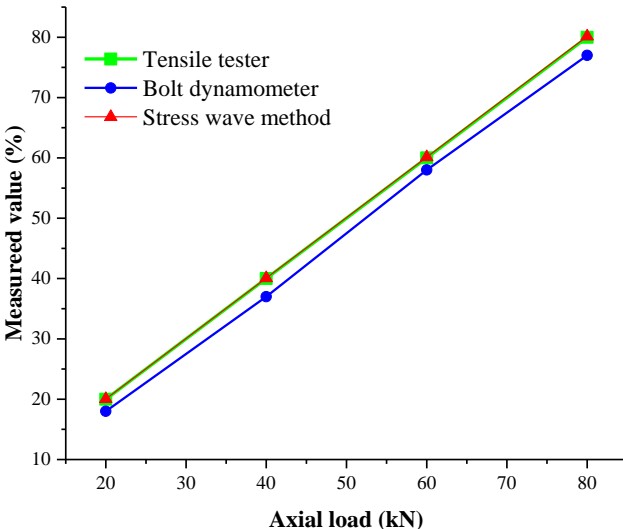

**Figure 15.** Comparison graph of three kinds of axial load measurement results.

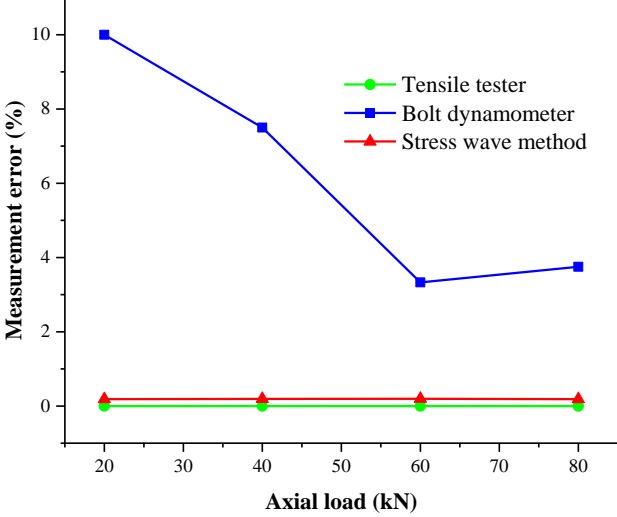

**Figure 16.** Comparison curve of three kinds of axial load measurement error.

Figure 15 shows the measurement results of the rock bolt axial load based on the stress wave time-domain amplitude method used in this study are closer to the data of the rock bolt tension machine, while the results measured by the traditional rock bolt dynamometer deviate from the expected experimental target value by 2 kN or 3 kN. That is because the rock bolt dynamometer acts on the pressure sensor by pulling, which is a more sensitive instrument to pressure. Meanwhile, the rock bolt force gauge is a pressure-sensitive precision instrument that is sensitive to temperature, humidity, air pressure, the collision of the measured object, and vibration. The output signal is easy to deviate, and then causes uncertainty in the test results. The measurement method used in this paper is to detect the stress wave signal, which excludes the uncertainties when using the rock bolt force gauge and thus reduces the influence of the interference signal after the signal processing in the later stage of the test.

Figure 16 shows that the relative errors measured by the rock bolt dynamometer and the stress-wave time-domain amplitude-based method were 3.33% and 0.185%, respectively. The error obtained by the rock bolt dynamometer was more prominent when the axial load is small. The testing error of the rock bolt dynamometer showed a tendency to decrease significantly first and then to increase slightly. The absolute error of the stress-wave time-domain amplitude-based method is much smaller than that of the traditional rock bolt

dynamometer method as the test load increases, i.e., the test results by the stress-wave method are more accurate. The error rate of the stress-wave time-domain amplitude method used in this paper can be effectively controlled in a very low stress range and is more stable than the traditional testing method, which verifies that the method can be used for determining the axial load of rock bolts and has significant advantages over the traditional nondestructive testing of rock bolts.

## 5. Conclusions

To investigate the influence of axial load on the characteristics of stress waves in rock bolts and verify the effectiveness of the NDT method of rock bolts based on the stress wave method, the stress waves were excited by applying an axial load to the anchor rods using a custom experimental bench. The stress wave signals were analyzed by various signal processing methods. Based on the work completed, the following conclusions can be drawn:

(1) The collected excitation–emission stress wave signal can be successfully filtered by using VMD decomposition and FFT low-pass filter. The analysis of the time domain characteristic parameters showed that the decay rate of the first cycle of the stress waveform in the time domain changed with the increase in the axial load; the processing of the frequency domain data using the FFT low-pass filter showed that the peak amplitude and shape of the stress waveform in the frequency domain were also affected by the axial load.

(2) For an intact and well-compacted rock bolt, the changes in axial load on the stress waveform signal characteristics of the rock bolt cannot be directly reflected from the time domain waveform and Fourier frequency domain and should be analyzed by combining multiple signal processing methods. The analysis results show that with the increase in axial load, the stress wave characteristics of the rock bolt change, and the time domain amplitude of the stress wave gradually decreases.

(3) The axial load of the rock bolt was determined by a dynamometer test and compared with the load value of the tensile machine to verify the nondestructive testing method based on the stress wave method used in this paper. The error of the method used in this paper is much smaller than that of the traditional anchor load test method, and the test results are more stable and were further applied to determine the working load of anchor rods.

**Author Contributions:** C.L. and X.X. designed the experiments; X.X. and X.G. carried out the experiments; X.C., G.L., J.B., B.N., Z.Z. and B.Z. processed the experiment's data. X.X. and C.L. analyzed the experimental results; X.X. and R.F. wrote the manuscript. All authors have read and agreed to the published version of the manuscript.

**Funding:** This work was supported by the National Natural Science Foundation of China (52174103), Anhui Provincial Natural Science Foundation (2008085ME147) and open foundation from the Key Laboratory of Coal Mine Safety and Efficiently Caving of Ministry of Education (JYBSYS2018102).

**Institutional Review Board Statement:** Not applicable.

**Informed Consent Statement:** Not applicable.

**Data Availability Statement:** Not applicable.

**Conflicts of Interest:** The authors declare that they have no known competing financial interest or personal relationships that could have appeared to influence the work reported in this paper.

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
