# Peer review of "Experimental Study of the Effect of Axial Load on Stress Wave Characteristics of Rock Bolts Using a Non-Destructive Testing Method"

_sustainability, doi:10.3390/su14159773_

Round 1

Reviewer 1 Report

The purpose of this paper is to verify the effectiveness of Non-destructive testing (NDT) of bolt using stress waves. In the test, the VMD decomposition method and Hilbert-Huang signal processing method were used to filter and analyze the stress wave signals. By studying the influence of different axial loads on stress waves, it was proved that the change of the amplitude of stress waves in time domain was correlated with the change of the amplitude of the load on rock bolt during elastic deformation. Furthermore, by comparing with the measured data, it is verified that the time-domain amplitude method of stress wave inversion is accurate and reliable to calculate the axial load of bolt. In a word, the research topic of this paper has practical value, rich content and clear logic progression, and lays a solid foundation for the development of rock bolt nondestructive testing technology in the future. However, the following modifications should be made:

1. The layout of the picture in Figure 1 is not reasonable. It is suggested to make it tabular and put the English number of the small picture under the picture. Other pictures are also recommended to check by themselves.

2. There is a large blank in line 140 and it is recommended to adjust the overall text of the article.

3. The values in Table 1 in line 161 are not all added in units. It is recommended to check whether there is any omission.

4. Equation (2) in line 223 suggests to prove the feasibility of introducing relevant factors by citing literature or norms.

5. The small title spacing of line 428 picture 14 is too large, it is suggested to adjust.

6. The topic is not sufficiently general and does not reflect this part of the paper ' stress wave method can be applied to nondestructive testing of rock bolts '. It is suggested to strengthen..

Author Response

Dear Editor and Reviewers,

Thank you for kindly processing our manuscript titled “Experimental Study of the Effect of Axial Load on the Stress Wave Characteristics of Rock Bolts” (Manuscript ID: sustainability-1841124). We greatly appreciate the insightful commons made by the reviewers, which are very helpful for improving our manuscript. We have made revisions according to the reviewers’ comments and please see our detailed responses below. All changes have been marked in the revised manuscript with yellow background. We hope that the revised version will meet the requirements of the journal and get your approval.

Reviewer 2 Report

This paper presents an experimental study on the effect of axial load on the stress wave characteristics of rock bolts. The variational mode decomposition (VMD) method and the Hilbert-Huang transform (HHT) signal processing method were mainly used to filter and analyze the stress wave signal. The results of this study have a certain reference value. However, some general corrections and weak points should be mentioned. The following comments are suggested:

1.         Abstract could be more concise.

2.         The purpose of signal processing and spectral analysis is to describe the spectral content of a signal over time, so that the energy or intensity of the signal can be represented in both time and spectrum. The author should emphasize this concept in the text.

3.         In Figure 2, there are two curves, one in blue and one in red. The authors should describe the differences and describe the legend.

4.         The content of subsection 3.2 can be more concise.

5.         The goal of VMD is to decompose a real-valued input signal into a discrete number of sub-signals that have a specific sparsity in reproducing the input signal. The sparsity is reflected in the bandwidth of the spectral domain. In other words, it is assumed that each sub-signal is mainly around the center frequency, which will be determined with the decomposition. The authors should emphasize this concept in the text.

6.         In Figure 5, the numbering of each sub-figure is wrong and should be corrected. In addition, some subplots lack titles for their vertical coordinates.

7.         The Hilbert-Huang transform (HHT) signal processing method is an algorithm applied to data. The authors should briefly describe the steps of its calculus.

8.     In  15, the authors should explain the trend of the measurement error using the rock bolt dynamometer method.

Author Response

(The authors gave the same response as above.)

Reviewer 3 Report

The authors verified the effectiveness of the stress wave anchor nondestructive testing method and solved the limitations of the traditional mine anchor testing, which is a very interesting study and worth investigating due to the tunnelling support issues in mining industry. The experimental methodology adopted in this study is novel and some insightful findings have been reported. However, before this manuscript can be considered for publication, some minor revisions are suggested:

1.      Section 4, please provide more explanations to describe the experimental results for the validation tests.

2.      The conclusion section should be simplied and expressed as concisely as possible.

3.      There are no descriptions regarding the subfigures (a) and (b) in Fig. 1 in the text. Please add corresponding statements for the customized experimental apparatus used in the study.

4.      The presentation of the testing scheme in Section 2.2 is not clear, so please add explanations about the operation steps of the test.

5.      The description of the change of stress waveform caused by the increase of axial load given after Figure 5 should be rephrased to make it clearer.

6.      Figure 8, more descriptions are needed to explain the reasons for the selection of the modal components.

7.      The formatting of the tables in the article is inconsistent, e.g., the width of the border in Table 2 is different from other tables, and the formatting in Tables 3, 4, 5, and 6 needs to be adjusted.

Author Response

(The authors gave the same response as above.)
